# The Problem of Filling a Spherical Cavity in an Aqueous Solution of Polymers

**DOI:** 10.3390/polym14204259

**Published:** 2022-10-11

**Authors:** Oxana A. Frolovskaya, Vladislav V. Pukhnachev

**Affiliations:** Lavrentyev Institute of Hydrodynamics SB RAS, Novosibirsk State University, Novosibirsk 630090, Russia

**Keywords:** dilute polymer solution, bubble cavity, relaxation viscosity, second-grade fluid

## Abstract

The problem of filling a spherical cavity in a liquid has attracted the attention of many authors. The study of bubble behavior in liquid allows to estimate the consequences of cavitation processes, which can lead to the intensive destruction of the material surface. Regarding this connection, it becomes necessary to study the influence of impurities, including polymeric additives on the strengthening or suppression of cavitation. In this paper, this problem is considered in three models of a relaxing fluid. It is shown that for all models, the cavity filling time is finite if the surface tension is not equal to zero. This result was previously established for the cases of ideal and viscous fluids. However, the relaxation factor can significantly change the flow pattern by slowing down the filling process and lowering the level of energy accumulation during the bubble collapse.

## 1. Introduction

The addition of a small amount of a polymer to water practically does not change the viscosity, density, thermal conductivity, and other properties of the solution. However, the liquid acquires relaxation properties, which leads to a sharp decrease in resistance when the solution moves in the pipe [1]. This publication gave rise to a large number of experimental studies of this phenomenon, which are reviewed in [2]. Various aspects of the dynamics of aqueous polymer solutions are discussed in a Special Issue of the Processes Journal [3].

The appearance of bubbles in a liquid, including a polymer solution, is inevitable. In particular, this is due to the possible appearance and suppression of bubble cavitation. The study of the motion of the aqueous solutions of polymers is important because the addition of soluble polymers to the liquid increases the stability threshold of the flow at high velocities. In addition, biopolymers also have relaxation properties. It is natural to expect that the relaxation properties of polymer solutions manifest themselves most noticeably in essentially nonstationary motions. An example of such a movement is the process of filling a spherical cavity in an initially at-rest liquid under the action of external pressure. The problem of filling a spherical bubble in an inviscid incompressible fluid was considered by Rayleigh [4]. In particular, he obtained that the rate of the cavity surface at the end of the filling increases without bound as $a^{-3/2}$, where a is the cavity radius. This can be interpreted as an unlimited energy cumulation in the process of focusing [5].

The introduction of viscosity and capillary forces into the game qualitatively changes the situation [6,7,8,9,10]. It turned out that there exists a critical Reynolds number Re∗ (depending on the Weber number), such that at Re>Re∗, the rate in the final stage of the focusing process grows as $a^{-3/2},$ similar to the case of an inviscid fluid. If Re<Re∗, then at a→0, the bubble surface rate tends to a finite limit $u_{*}=-\sigma /2\rho \nu ,$ where ρ is the liquid density, ν is the kinematic viscosity, and σ is the surface tension coefficient. It can be stated that when Re<Re∗, the action of viscous forces eliminates the cumulation of energy. We note that in all the cases mentioned above, at σ>0, the filling of the cavity occurs in a finite time. If σ=0, then at Re<Re∗≈8.4, the filling time becomes infinite.

In more complex fluid models, the problem under discussion was considered in [11,12,13,14]. In [11], the existence of three modes of motion of the cavity boundary in an incompressible viscoelastic Maxwell medium is established. It is shown that the filling process can be both monotonous and oscillatory. Bubble dynamics in a compressible viscoelastic liquid is studied in [12]. The collapse of a spherical bubble in viscous incompressible fluid with nonlinear viscosity is investigated in [13,14]. The addition of polyacrylamide (PAM) with the formula (-CH_2_CHCONH_2_-) to water affects the moment of occurrence of cavitation, i.e., a decrease in the number of cavitation is observed [14]. It turned out that the effect of polymers on single bubble dynamics is very small [15,16,17]. In the experimental work [18], the behavior of a bubble near a solid wall is studied and it is concluded that polymer additives do not significantly affect the bubble dynamics. A study on the dynamics of a spherical gas bubble in an incompressible power-law non-Newtonian fluid [19] showed that for some certain indices, there is no energy concentration at all. An experimental study [20] has shown that polymeric additives in water affect the cavitation phenomena and reduce the critical cavitation number. For instance, polymeric additives suppress the erosion of materials during cavitation developing in a flow [21]. The formation and collapse of a vapor bubble in the aqueous solution of PAM is studied experimentally in [22]. The experiments did not reveal a noticeable deceleration of bubble collapse. Our goal is to consider this problem for the case of a relaxing fluid.

## 2. Mathematical Models in the Dynamics of Polymer Solution

The viscous fluid model cannot always be applied to describe the motion of real fluids. Therefore, there is a need to complicate the model. There is no single point of view on how to complicate the classical viscous fluid models for studying the flow with small polymer additives.

Further, the fluid is assumed to be incompressible, and its density ρ, viscosity ν, and surface tension coefficient σ are assumed to be constant.

A model for the motion of polymer solutions, considering their relaxation properties, was proposed by Voitkunskii, Amfilokhiev, and Pavlovskii [23]. The authors used the ideas of the hereditary theory of viscoelasticity [24,25]. In their model, the relationship between the shear stress tensor P and strain rate tensor D has the form
(1)$$P=-pI+2\rho \nu D+2\frac{\rho \kappa }{\theta }\int_{0}^{t}\exp\left(\frac{z-t}{\theta }\right)\frac{dD}{dz}dz.$$
here, p is the pressure, I is the unit tensor, θ is the shear stress relaxation time, and κ is the normalized relaxation viscosity [26]. The symbol d/dt=∂/∂t+v⋅∇, where v is the velocity vector, means the operator of total differentiation with respect to time t. The parameters θ and κ are also considered constant. The value κ has the dimension cm^2^. The relaxation time of an aqueous solution of PAM with a concentration of $10^{-2}$ percent is of the order of $10^{-4}$ s. In the case of the relaxation viscosity coefficient, the authors of the model did not provide its characteristic values, though one can assume that they are sufficiently small. In [26], we discussed the possibility of the experimental determination of this parameter.

The alternative model is the Oldroyd-B model [27]. This model was used in [22] for a theoretical study of the growth and collapse of relatively small bubbles. It has been shown that polymeric additives, in principle, can reduce the cavitation erosion of material.

The system of equations of motion of an incompressible continuous medium has the form
$$\rho \frac{dv}{dt}=\mathrm{Div}P,\;\mathrm{div}v=0.$$

The first equation is the momentum equation for a continuous medium obeying the Cauchy stress principle, and the second one is the continuity equation for an incompressible continuous medium. The divergence of a tensor is a vector whose components are the divergences of the row vectors that form the given tensor.

Substituting expression P (1), here, we obtain:(2)$$\frac{dv}{dt}=-\frac{1}{\rho }\nabla p+\nu \Delta v+\frac{\kappa }{\theta }\int_{0}^{t}\exp\left(\frac{z-t}{\theta }\right)\frac{d\Delta v}{dz}dz,\;\mathrm{div}v=0.$$

Using the smallness of the parameter, θ, Pavlovskii [28] simplified Equation (2), restricting himself to the main term of the integral expression at θ→0,
(3)$$P=-pI+2\rho \nu D+2\rho \kappa \frac{dD}{dt}.$$

As a result, a system of equations is written as follows:(4)$$\frac{dv}{dt}=-\frac{1}{\rho }\nabla p+\nu \Delta v+\kappa \frac{d\Delta v}{dt},\;\mathrm{div}v=0.$$

In [26], an exact reduction of integro-differential Equation (2) to a system of differential equations was found,
(5)$$\theta \frac{\partial }{\partial t}\frac{dv}{dt}+\frac{dv}{dt}=-\frac{\theta }{\rho }\frac{\partial \nabla p}{\partial t}-\frac{1}{\rho }\nabla p+\theta \nu \frac{\partial \Delta v}{\partial t}+\nu \Delta v+\kappa \frac{d\Delta v}{dt},\mathrm{div}v=0,$$
and some exact solutions of this system are found.

One more possible modification of the model of motion of dilute aqueous polymer solutions is the introduction of an objective time derivative of the tensor D [29,30],
$$\frac{\tilde{d}D}{dt}=\frac{\partial D}{\partial t}+(v\cdot \nabla )D+DW-WD.$$
here, W is the antisymmetric part of the tensor ∇v. In this case, instead of system (4), we obtain a system of second-grade fluid equations [31,32]:(6)$$\frac{dv}{dt}=-\frac{1}{\rho }\nabla p+\nu \Delta v+2\kappa \mathrm{Div}\left(\frac{\tilde{d}D}{dt}\right),\;\mathrm{div}v=0.$$

The replacement of the total derivative of the tensor D with respect to time by its objective derivative is caused by the necessity to provide the rheological relation connecting the tensors P and D tensor-invariant form [29],
(7)$$P=-pI+2\rho \nu D+2\rho \kappa \frac{\tilde{d}D}{dt}.$$

The medium behavior law given by relation (7) does not change after transfer from the original coordinate system to another system rotating relative to it with an arbitrary angular velocity [30]. The relation (3) does not have this property. This allows us to consider system (6) to be more preferable for modeling the motion of aqueous polymer solutions than system (4). The mathematical theory of second-grade fluid was developed in [33,34] (see also the literature cited there).

It is important to note that the second-grade fluid equations allow an “a priori” estimate for the initial boundary value problem solution. This makes it possible to prove existence and uniqueness theorems for the solution of this problem without restrictions on the time interval and the norm of the initial data. For the Pavlovskii model, this fact does not hold.

Note that in the case of potential flows, where W=0, systems (4) and (6) coincide. This particular situation is considered in the present work.

## 3. Problem Formulation and Similarity Criteria

Let us assume that the motion is spherically symmetric and denote by u(r,t) the radial component of the fluid velocity, where r is the spherical radius. The origin of coordinates of the spherical system (r, θ, φ) is chosen at the center of the bubble which is at rest. The continuity equation (the second equation of system (2)) in spherical coordinates takes the form
(8)$$\frac{\partial u}{\partial r}+\frac{2u}{r}=0.$$

In order to satisfy the continuity Equation (8), the velocity component *u* is given as
(9)$$u(r,t)=\frac{f(t)}{r^{2}},$$
where *r*(*t*) is the bubble radius at time *t*. The motion described by the velocity field of the form (9) is potential. Therefore, Δv=0, and in the momentum equation (the first equation of system (2)), the last two terms in the right part vanish. The projection of the momentum equation onto the spherical coordinate axis r has the form
(10)$$\frac{\partial u}{\partial t}+u\frac{\partial u}{\partial r}=-\frac{1}{\rho }\frac{\partial p}{\partial r}.$$

Integrating Equation (10) with the condition $p\to p_{0}$ at r→∞ gives
(11)$$\frac{p-p_{0}}{\rho }=\frac{f\prime }{r}-\frac{f^{2}}{2r^{4}},$$
where prime denotes differentiation with respect to *t*. We see that the ordinary and relaxation viscosities do not enter into the momentum equation. However, both of these factors manifest themselves in the dynamic condition at the free boundary, to the derivation of which we turn. The expressions for non-zero elements of the tensor dD/dz in formula (1) read
(12)$$\frac{dD_{rr}}{dz}=-\frac{2f^{\prime }(z)}{r^{3}}+\frac{6f^{2}(z)}{r^{6}},\;\frac{dD_{\theta \theta }}{dz}=\frac{dD_{\varphi \varphi }}{dz}=\frac{f^{\prime }(z)}{r^{3}}-\frac{3f^{2}(z)}{r^{6}}.$$

However, only the term $dD_{rr}/dz$ contributes to the normal stress on a sphere of radius *r*. From Equality (1), we obtain the formula for the normal stress:(13)$$p_{nn}=-p-4\rho \nu \frac{f}{r^{3}}+\frac{2\rho \kappa }{\theta }\int_{0}^{t}\exp\left(\frac{z-t}{\theta }\right)\left(\frac{6f^{2}(z)}{r^{6}}-\frac{2f^{\prime }(z)}{r^{3}}\right)dz.$$

At the free boundary, the dynamic condition is written as follows
(14)$$p_{nn}=\frac{2\sigma }{s}\;\;\mathrm{at}\;r=s,$$
where s(t) is the cavity radius. Assuming r=s in (11)–(14) and excluding $p_{nn}$ from Equalities (13), (14), we obtain
(15)$$\frac{2\sigma }{\rho s}=-\frac{p_{0}}{\rho }-\frac{f^{\prime }}{s}+\frac{f^{2}}{2s^{4}}-4\nu \frac{f}{s^{3}}+\frac{2\kappa }{\theta }\int_{0}^{t}\exp\left(\frac{z-t}{\theta }\right)\left(\frac{6f^{2}(z)}{s^{6}}-\frac{2f^{\prime }(z)}{s^{3}}\right)dz.$$

Equation (15) can be reduced to a differential equation. It follows from the kinematic boundary condition at the free boundary
(16)$$\frac{ds}{dt}=u(s,t),$$
that $f(t)=s^{2}(t)s^{\prime }(t).$ First, we substitute the last expression into Equation (15),
(17)$$\frac{p_{0}}{\rho }+ss^{\prime \prime }+\frac{3{s^{\prime }}^{2}}{2}+4\nu \frac{s^{\prime }}{s}+\frac{2\sigma }{\rho s}+\frac{4\kappa }{\theta }\int_{0}^{t}\exp\left(\frac{z-t}{\theta }\right)\left(\frac{s^{\prime \prime }(z)}{s(z)}-\frac{s^{\prime }{\left(z\right)}^{2}}{s{\left(z\right)}^{2}}\right)dz=0.$$

Then, we differentiate Equality (17) and multiply the result by θ
$$\theta \left(ss^{\prime \prime \prime }+\frac{4}{s}(ss^{\prime }+\nu )s^{\prime \prime }-\frac{4\nu {s^{\prime }}^{2}}{s^{2}}-\frac{2\sigma s^{\prime }}{\rho s^{2}}\right)+\frac{4\kappa s^{\prime \prime }}{s}-\frac{4\kappa {s^{\prime }}^{2}}{s^{2}}+\frac{4\kappa }{\theta }\int_{0}^{t}\exp\left(\frac{z-t}{\theta }\right)\left(\frac{s^{\prime \prime }(z)}{s(z)}-\frac{s^{\prime }{\left(z\right)}^{2}}{s{\left(z\right)}^{2}}\right)dz=0.$$

From here and (17), we get
(18)$$2\theta \left(s^{3}s^{\prime \prime \prime }+4s(ss^{\prime }+\nu )s^{\prime \prime }-4\nu {s^{\prime }}^{2}-\frac{2\sigma }{\rho }s^{\prime }\right)+2s(s^{2}+4\kappa )s^{\prime \prime }+(3s^{2}-8\kappa ){s^{\prime }}^{2}+8\nu ss^{\prime }+\frac{4\sigma }{\rho }s+\frac{2p_{0}}{\rho }s^{2}=0.$$

Assuming that θ→0 in Equation (18), we obtain the equation for s in the second-grade fluid and the Pavlovskii models,
(19)$$2s(s^{2}+4\kappa )s^{\prime \prime }+(3s^{2}-8\kappa ){s^{\prime }}^{2}+8\nu ss^{\prime }+\frac{4\sigma }{\rho }s+\frac{2p_{0}}{\rho }s^{2}=0.$$

The initial conditions are determined from the assumption of the emergence of motion from a state of rest under the action of a pressure impulse. At the initial moment, the bubble radius is given and equal to $r_{0}$, and the initial velocity is equal to zero. For Equation (19), these conditions are
(20)$$s(0)=r_{0},\;s^{\prime }(0)=0.$$

The set of initial conditions for Equation (18) is as follows
(21)$$s(0)=r_{0},\;s^{\prime }(0)=0,\;s^{\prime \prime }(0)=-\frac{p_{0}}{\rho r_{0}}-\frac{2\sigma }{\rho r_{0}^{2}}.$$

Equation (18) include four parameters ρ,ν,κ,θ characterizing the medium. Two more dimensional parameters of the problem are the initial radius of the bubble $r_{0}$ and the pressure far from it, which is associated with atmospheric pressure, $p_{0}$. From these parameters, four dimensionless combinations can be made, which are the similarity criteria in our problem
Re=r0νp0ρ, β=σνρp0, γ=κp0ρν2, τ=θp0ρν.

For a given $p_{0}=101325$ Pa, the quantities β, γ and τ are determined only by the properties of the solution. At a temperature of 293 K for water ρ=0.998 g/cm^3^, ν=0.01 cm^2^/s, σ=72.8 dyn/cm, then β=22.89. When calculating the parameter τ, considering the order of the relaxation time of an aqueous solution of PAM with a concentration of approximately $10^{-2}$%, specified in [23], $10^{-4}$ s is taken, which gives $\tau =10^{3}$. As for the parameter γ, it is more difficult to indicate its characteristic values since there are no direct experiments from which the value of the quantity κ is found. However, even at $\kappa =10^{-4}$ cm^2^, the values of this parameter will be very large, $\gamma =10^{5}$. The last of the four similarity criteria is the Reynolds number. This parameter is at our disposal. However, there are physical limitations from above on the value of Re, which are related to the fact that regarding large bubble diameters, it is difficult to ensure the sphericity of its shape. Assuming $r_{0}=0.2$ cm, we find Re=6372, and for $r_{0}=0.01$ cm, we get Re=318.63.

The problem in a dimensionless form is formulated by choosing the following values for normalization: $\sqrt{p_{0}/\rho }/\nu$ for the radius of the bubble, and $p_{0}/(\rho \nu )$ for time. The relations (18)–(21) in dimensionless variables are written as (the same letters are used to denote non-dimensional variables)
(22)$$2\tau \left(s^{3}s^{\prime \prime \prime }+4s(ss^{\prime }+1)s^{\prime \prime }-4{s^{\prime }}^{2}-2\beta s^{\prime }\right)+2s(s^{2}+4\gamma )s^{\prime \prime }+(3s^{2}-8\gamma ){s^{\prime }}^{2}+8ss^{\prime }+4\beta s+2s^{2}=0.$$
(23)s(0)=Re, s′(0)=0, s″(0)=−1Re−2βRe2.
(24)$$2s(s^{2}+4\gamma )s^{\prime \prime }+(3s^{2}-8\gamma ){s^{\prime }}^{2}+8ss^{\prime }+4\beta s+2s^{2}=0.$$
(25)s(0)=Re, s′(0)=0.

The differential Equations (22) and (24) are the ordinary differential equations. For numerical implementation, the equations are written as a system of first-order ordinary differential equations. The initial-value problems (22), (23) and (24), (25) are solved using the fourth-order Runge-Kutta method. The presence of a large parameter γ in front of the highest derivative does not complicate but simplifies calculations. In the following, the results of the numerical solution are given in a dimensionless form.

## 4. Hereditary Model

The bubble dynamics in the hereditary model is described in terms of the solution of the Cauchy problem (22), (23). This problem contains four dimensionless parameters: τ, β, γ, Re. Let us first consider the case of large values of the parameter τ at fixed values of the other parameters.

Let us divide both parts of the equation
$$\tau \left(s^{3}s^{\prime \prime \prime }+4s(ss^{\prime }+1)s^{\prime \prime }-4s^{\prime 2}-2\beta s^{\prime }\right)+s(4\gamma +s^{2})s^{\prime \prime }+(-4\gamma +3s^{2}/2){s\prime }^{2}+2(2s^{\prime }+\beta )s+s^{2}=0$$
by τ and pass to the limit τ→∞. The resulting equation
$$s^{3}s^{\prime \prime \prime }+4s(ss^{\prime }+1)s^{\prime \prime }-4{s\prime }^{2}-2\beta s^{\prime }=0$$
after dividing by $s^{2}$ allows integration,
(26)$$s^{2}s^{\prime \prime }+3s{s^{\prime }}^{2}/2+4s^{\prime }+2\beta -qs=0,$$
where q is the constant of integration, which will be chosen below. The transition in Equation (26) to the independent variable s, and the new desired function $b(s)=s^{\prime }(t)$, gives
(27)$$\frac{db}{ds}=-\frac{4b+(3sb^{2}/2)-qs+2\beta }{s^{2}b}.$$

Equation (27) has a single singular point, s=0, b=−β/2. Passing in (27) to the new desired function, c=b+β/2, we arrive at the equation
$$\frac{dc}{ds}=\frac{8c+3sc^{2}-6\beta sc+2s(3\beta^{2}/8-q)}{s^{2}(\beta /2-c)}.$$

The singular point here is the origin of coordinates on the plane s, c. Taking $8q=3\beta^{2}$, we transform the last equation to the following form:$$\frac{dc}{ds}=\frac{8c}{\beta s^{2}}-\frac{6c}{s}+\frac{2c}{\beta }\frac{dc}{ds}+\frac{3c^{2}}{\beta s}.$$

This equation, in turn, can be reduced to the integral equation
(28)$$c(s)=Ks^{-6}\exp(-8/\beta s)+s^{-6}\exp(-8/\beta s)\int_{0}^{s}\left(\frac{2c(z)}{\beta }\frac{dc(z)}{dz}+\frac{3c^{2}(z)}{\beta z}\right)\;dz,$$
where K=const. Equation (28) has the unique solution c(s), which is continuous and has the continuous derivative on segment [0,δ] if δ>0 is sufficiently small. This solution can be obtained in the limit of an iteration process. For small s, the asymptotics of the solution of Equation (28) is given by the relation $c=Ks^{-6}\exp(-8/\beta s)[1+o(1)].$ From here, we find:$$s^{\prime }\equiv b(s)=-\beta /2+Ks^{-6}\exp(-8/\beta s)[1+o(1)].$$

Positive solutions of this equation extremely quickly reach the asymptotics
s=A−βt/2, A=const>0.

The lifetime of the solution of Equation (26) is close to T=2A/β. The limiting value of the bubble boundary rate at t→T is −β/2. The dimensional value of this velocity is given by the formula $V^{*}=-\sigma /2\rho \nu$. It is remarkable that the same value was obtained by Galperin (see [10], pp. 121–124) in the problem of bubble flow in an ordinary viscous liquid if the Reynolds number is less than the critical value Re∗=Re∗(β). A similar behavior is demonstrated by the solution of problem (22), (23) at moderate values of the parameter τ.

The results of the numerical solution of the problem (22), (23) are presented in Figure 1 for $\beta =22.89,\tau =10^{3},\;\gamma =10^{5},$ and various values of the Reynolds number Re.

Recall that $\gamma =\kappa p_{0}/\rho \nu^{2}$. Assuming ν=1.48 cm^2^/s, ρ=1.26 g/cm^3^, which corresponds to glycerol at a temperature of 293 K and $\kappa =10^{-4}$ cm^2^, we find γ=3.67. By choosing the relaxation time $\theta =10^{-3}$ s, we get τ=54.34. For such values of the chosen parameters and β=0.0048, Re=4, we ensure the finiteness of the bubble boundary velocity at the moment of focusing. Figure 2 demonstrates the result of numerical solution of problem (22), (23) at Re=4,β=0.0048,γ=3.67, and τ=54.43.

Figure 3 shows the results of the numerical solution of problem (22), (23) at Re=5,β=0.0048,γ=5, and various values of the parameter τ.

The calculations show that as the parameter β (proportional to the surface tension coefficient) decreases, the bubble filling time increases. Figure 4 and Figure 5 demonstrate solutions *s*(*t*) of problem (22), (23) for $\tau =10^{3},\;\gamma =10^{5}$, and various values of the Reynolds number and the parameter β. One can see that the bubble filling time decreases when the surface tension β increases.

Consider the limiting case β=0, we rewrite Equation (22) as
(29)$$8ss^{\prime }+2s^{2}+8(\tau +\gamma )(ss^{\prime \prime }-{s^{\prime }}^{2})+s^{2}\left(2\tau (ss^{\prime \prime \prime }+4s^{\prime }s^{\prime \prime })+2ss^{\prime \prime }-3{s^{\prime }}^{2}\right)=0.$$

According to (23), the initial conditions for Equation (29) at β=0 have the form
(30)s(0)=Re, s′(0)=0, s″(0)=−1Re.

The asymptotics of the solution of the problem (29), (30) at t→∞ is looked for in the form
$$s=\sum_{n=1}^{\infty }s_{n}\exp(-nt/4).$$

If the value $s_{1}$ is known, the remaining constants $s_{n}\;(n=2,3,\ldots )$ are determined sequentially from recurrence relations. In particular, $s_{2}=0,\;s_{3}=(5\tau +2)s_{1}^{3}/64(\tau +\gamma -2).$ The results of the calculations performed at Re=10, τ=10, γ=100 are presented in Figure 6. Thus, at β=0, the cavity is filled in infinite time.

## 5. Pavlovskii Model

The process of filling a spherical cavity in Pavlovskii [28] and second-grade fluid [31] models is described in terms of solving the Cauchy problem (24), (25). Due to conditions (25), $s^{\prime \prime }(0)<0.$ Therefore, on some interval (0, l), the inequality $s^{\prime }(t)<0$ will be fulfilled. Let us show that this inequality holds over the whole interval [0, T] of existence of the solution of problem (24), (25). In fact, if it is not so, there is a value $\bar{t}$, such that $s^{\prime }(\bar{t})=0$, and the point $\bar{t}$ is a minimum point of the function s(t), such that $s^{\prime \prime }(\bar{t})\geq 0.$ If in this case $s(\bar{t})>0$, then we come to a contradiction with Equation (24). However, the possibility that the relations $s^{\prime }(\bar{t})=0,\;s(\bar{t})=0$ can be executed simultaneously is not excluded. This is the situation in our problem. To confirm this, it is necessary to study the behavior of small solutions of Equation (24).

Passing in Equation (24) to a new independent variable s and a new desired function a(s)=ds/dt, we obtain the equation on the phase plane s, a:(31)$$\frac{da}{ds}=\frac{(8\gamma -3s^{2})a^{2}-8sa-4\beta s-2s^{2}}{(8\gamma +2s^{2})sa}.$$

Equation (31) has a single singular point s=0, a=0. To study the behavior of its solutions near a singular point, let us substitute in Equation (31)
(32)$$a=-{\left(\beta s/\gamma \right)}^{1/2}b.$$

The function b(s) satisfies the equation
(33)$$\frac{db}{ds}=\frac{2\gamma (b^{2}-1)+4{\left(\gamma s/\beta \right)}^{1/2}b-\gamma s/\beta -2s^{2}b^{2}}{(4\gamma +s^{2})sb}.$$

The singular point s=0, b=1 of Equation (33) corresponds to the singular point s=a=0 of Equation (31). The asymptotics of the solution of this equation at s→0 has the form
(34)$$a=-{\left(\beta s/\gamma \right)}^{1/2}[1-2{\left(s/\gamma \beta \right)}^{1/2}+O(s\log s)],\;s\to 0.$$

Relations (32), (34) allow us to find the main term of the asymptotics of the function s(t) at t→T:(35)$$s=\beta {\left(T-t\right)}^{2}/4\gamma +o{\left(T-t\right)}^{2}.$$

Here, there is a qualitative difference between the final stages of the bubble collapse process in the hereditary model and in the Pavlovskii and the second-grade fluid models. If, in the first case, the rate of the free boundary at the moment of collapse is finite, then in the second case, this rate tends to zero at t→T.

Figure 7 shows the solutions *s*(*t*) of problem (24), (25) at β=22.89, $\gamma =10^{5}$, and various Reynolds numbers Re.

Figure 8 illustrates the solutions *s*(*t*) of problem (24), (25) at Re=318.63, β=22.89, and various values of the parameter γ.

Figure 9 and Figure 10 demonstrate solutions *s*(*t*) of problem (24), (25) for $\gamma =10^{5}$, and various values of the Reynolds numbers Re and the parameter β.

## 6. Pressure

Pressure distribution and the normal stress in the Pavlovskii and the second-grade fluid models in dimensionless variables are given by the formulae
(36)$$p(r,t)=1+\frac{f^{\prime }(t)}{r}-\frac{f^{2}(t)}{2r^{4}},\;f(t)=s^{2}(t)s^{\prime }(t)$$
and
(37)$$p_{nn}=-p-\frac{4f(t)}{r^{3}}+4\gamma \left(\frac{3f^{2}(t)}{r^{6}}-\frac{f^{\prime }(t)}{r^{3}}\right)$$
respectively. Figure 11, Figure 12 and Figure 13 illustrate the distributions of pressure in the liquid surrounding the bubble in the Pavlovskii and the second-grade fluid models at the time moment, when the bubble radius is less than the initial one by approximately 10 times for Re=318.63,β=22.89 and the various values of the parameter γ.

Note that as the parameter γ increases, the maximum pressure increases and then begins to decrease. From Figure 14, it can be seen that the plot for an ordinary liquid gives close values of the maximum pressure to those indicated in [35], where the problem for an ideal liquid was considered.

The more important characteristic is the normal stress. Figure 15 shows the normal stress in the fluid surrounding the bubble for Re=318.63,β=22.89, and the various values of the parameter γ and at different points in time. Here, we observe a non-monotonic character in the distribution of normal stresses depending on the parameter γ proportional to the relaxation viscosity coefficient.

## 7. Discussion

1. In solving the problem of filling a spherical cavity in the hereditary model, the velocity of its boundary is bounded at s→0, where s is the radius of the cavity. This property does not depend on the Reynolds number. As for the Pavlovskii and the second-grade fluid models, here, the velocity of a cavity boundary tends to zero as s→0. At the same time, in a Newtonian fluid, this asymptotic behavior is of the order of $s^{-3/2}$ if the Reynolds number exceeds the critical value. Thus, the phenomenon of energy accumulation in a relaxing liquid in the process of a cavity filling does not take place.

2. Regarding the cavity collapse time in solving problem (24), (25) increases with the Reynolds number. We have s′(0)=0, s″(0)=−(Re+2β)/(Re2+4γ) and s″(0)=O(Re−1) at Re→∞. Thus, an increase in the Reynolds number leads to a deceleration in the motion of the cavity boundary at the initial moment of time. Calculations show that this trend persists over time. In the case of a Newtonian fluid, s″(0)=−(Re+2β)/Re2. Here, the process of wicking the cavity slows down with increasing Re too, but the deceleration occurs faster, since there is no term 4γ in the denominator of the last formula, and in typical situations, the value γ is of the order $10^{5}$.

3. In the problem of cavity filling both in the hereditary model and in the Pavlovskii and second-grade fluid models, for some values of relaxation viscosity and shear stress relaxation time, there is no energy cumulation. The velocity of the bubble boundary remains finite as its radius tends to be zero. This regime is similar to that which occurs for a Newtonian fluid at subcritical Reynolds numbers.

4. It is of interest to see the pressure behavior. In the literature available to us, there are no plots of pressure and normal stress during the collapse of a cavity in a relaxing liquid. The exception is the work [12], but in it, the filling of the cavity occurs under the action of high pressures, which leads to the need to consider the compressibility of the liquid. The maximum pressure that occurs at a moment of time, close to the moment of collapse, reaches a value of the order of $10^{2}$. The maximum impulse pressure obtained in [12] is of the order of $10^{3}$. Note the dual role of relaxation. On the one hand, relaxation slows down the process of bubble collapse, and on the other hand, the pressure increases near the moment of collapse.

## Figures and Tables

**Figure 1 polymers-14-04259-f001:**
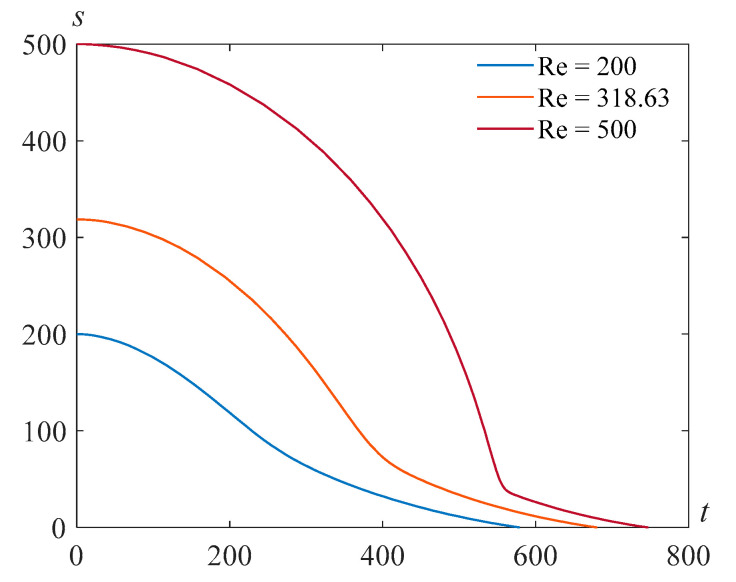
Cavity radius versus time for $\beta =22.89,\tau =10^{3},\;\gamma =10^{5},$ and various values of the Reynolds number Re.

**Figure 2 polymers-14-04259-f002:**
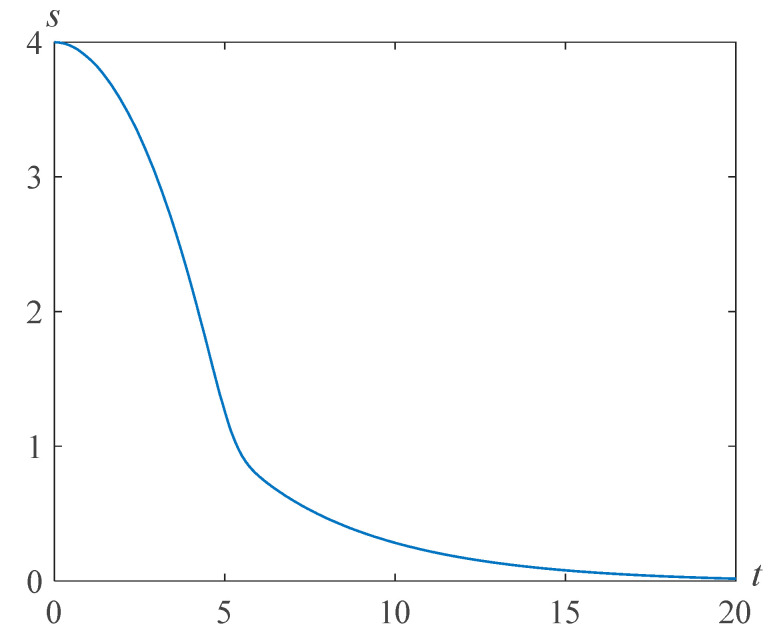
Cavity radius versus time for Re=4,β=0.0048,γ=3.67, and τ=54.43.

**Figure 3 polymers-14-04259-f003:**
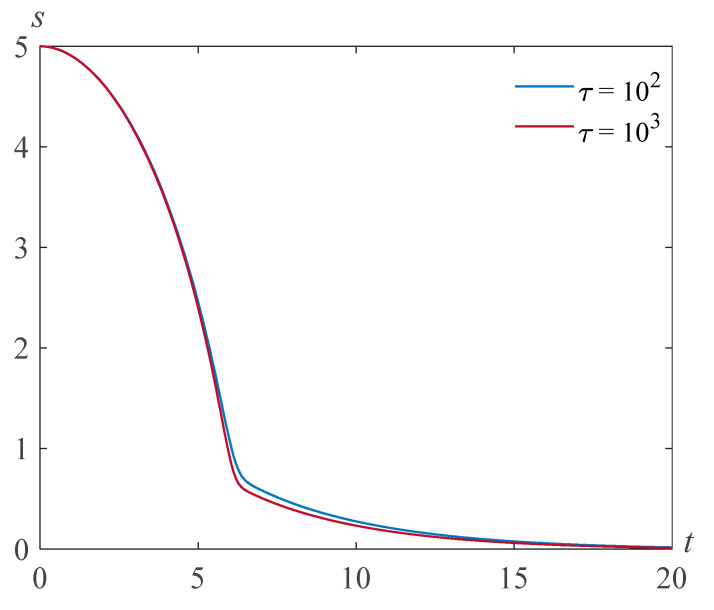
Cavity radius versus time for Re=5,β=0.0048,γ=5, and various values of the parameter τ.

**Figure 4 polymers-14-04259-f004:**
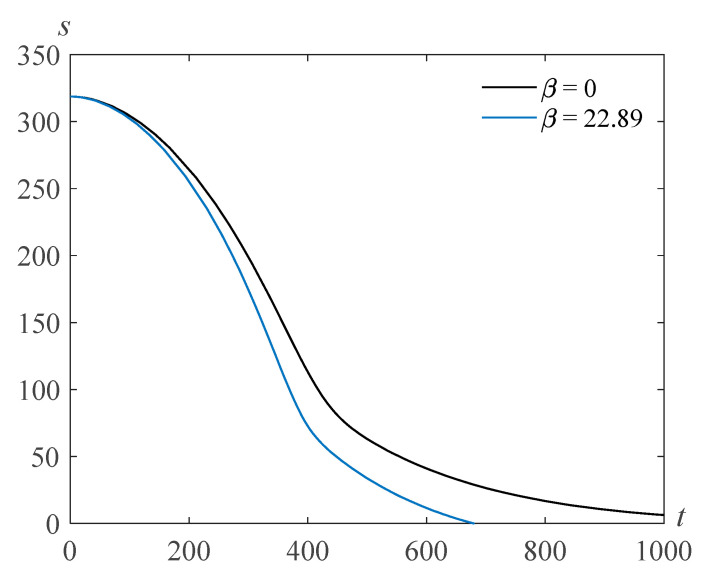
Cavity radius versus time for Re=318.63, τ=103, γ=105,  and various values of the parameter β.

**Figure 5 polymers-14-04259-f005:**
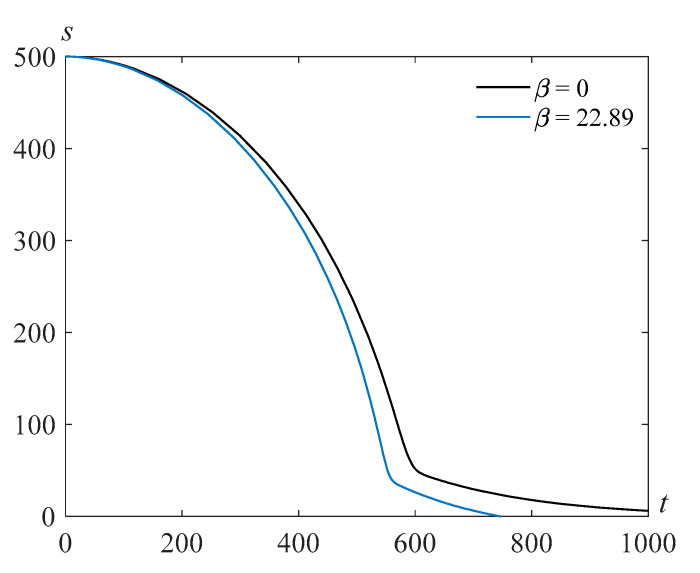
Cavity radius versus time at Re=500, τ=103, γ=105, and various values of the parameter β.

**Figure 6 polymers-14-04259-f006:**
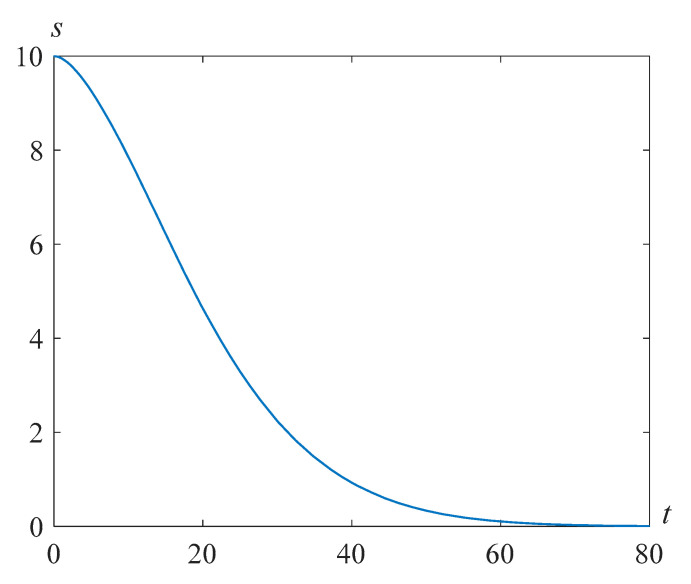
Cavity radius versus time at Re=10, τ=10, γ=100, and β=0..

**Figure 7 polymers-14-04259-f007:**
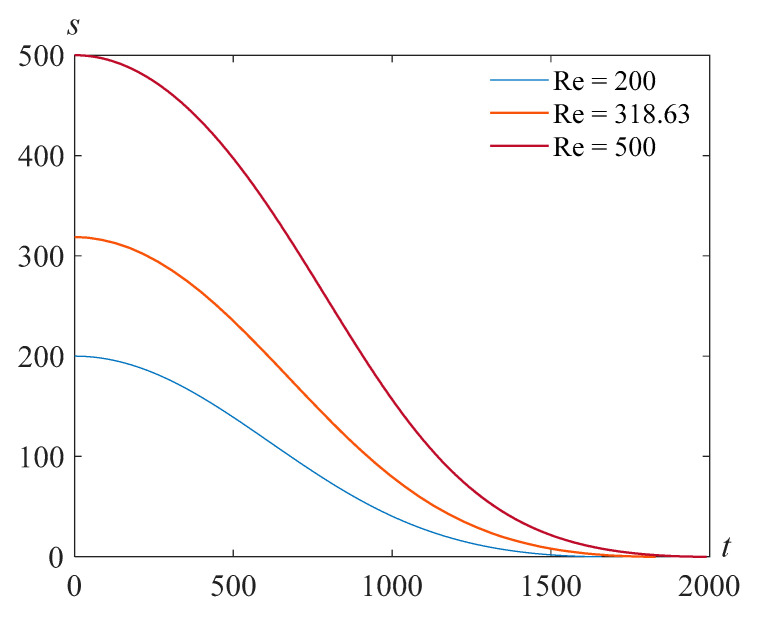
Cavity radius versus time for $\beta =22.89,\;\gamma =10^{5},$ and various values of the Reynolds number Re.

**Figure 8 polymers-14-04259-f008:**
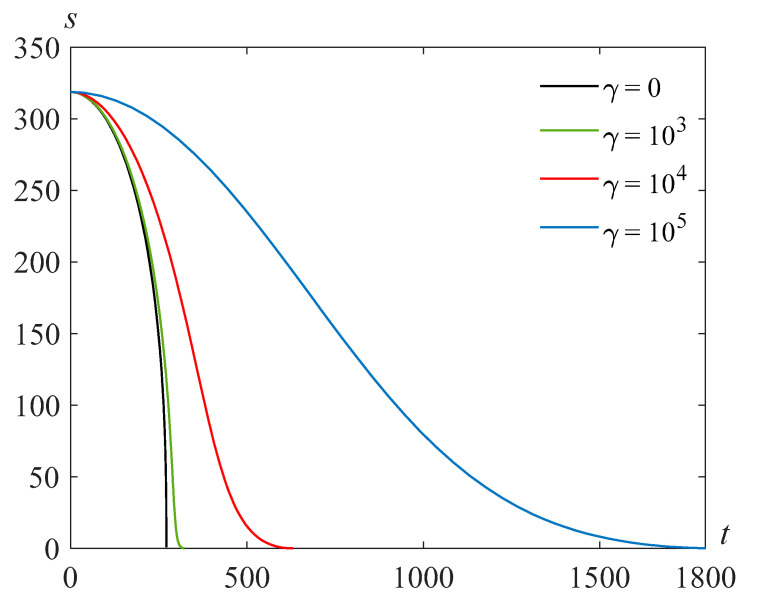
Cavity radius versus time for Re=318.63, β=22.89, and various values of the parameter γ.

**Figure 9 polymers-14-04259-f009:**
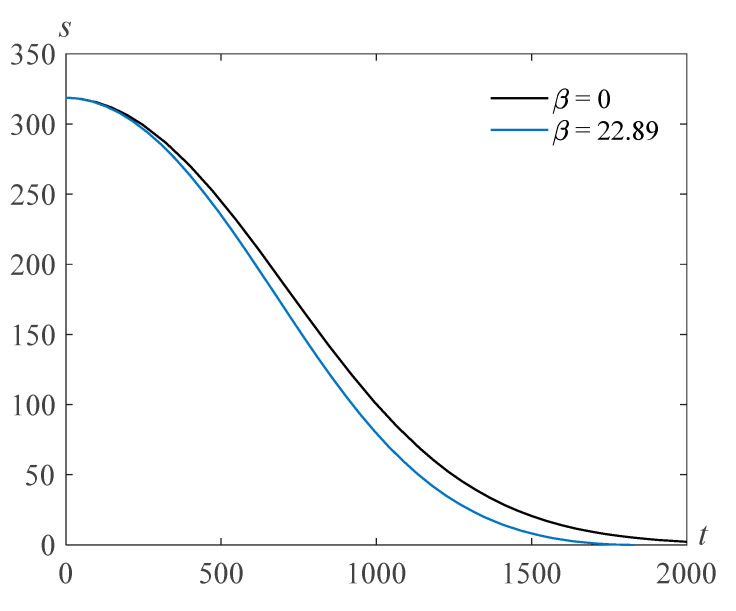
Cavity radius versus time for Re=318.63, γ=105, and various values of the parameter β.

**Figure 10 polymers-14-04259-f010:**
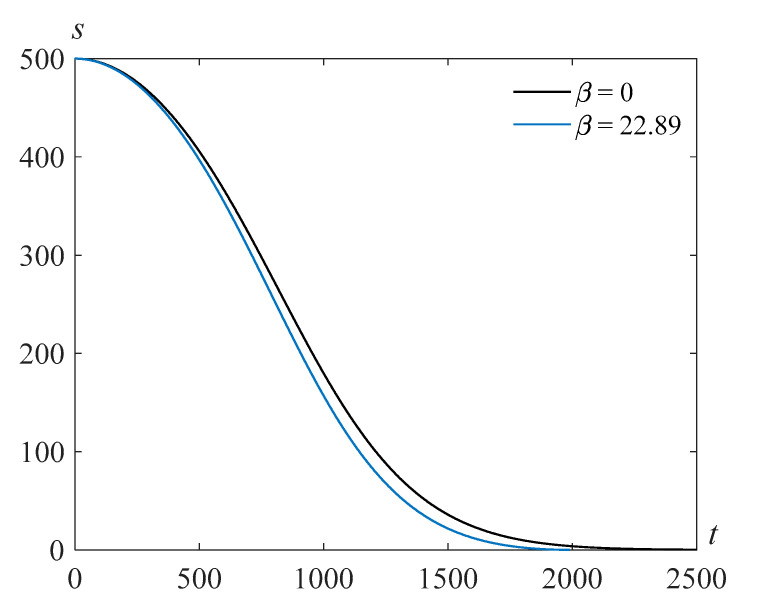
Cavity radius versus time for Re=500, γ=105, and various values of the parameter β.

**Figure 11 polymers-14-04259-f011:**
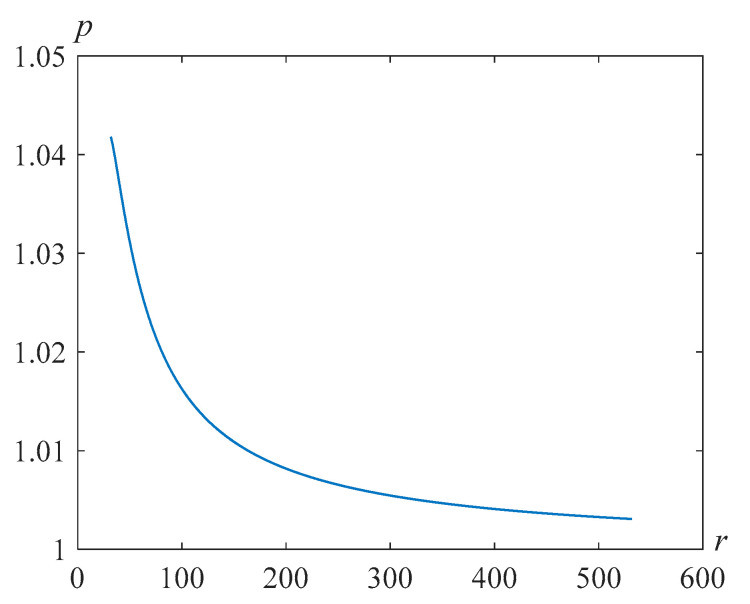
Pressure distribution in the fluid surrounding the bubble for Re=318.63,β=22.89,γ=105, and at the time t=1247.

**Figure 12 polymers-14-04259-f012:**
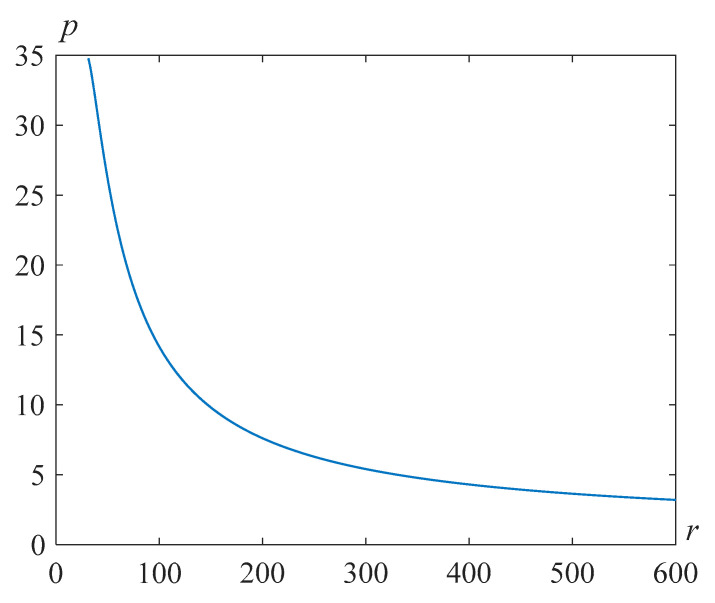
Pressure distribution in the fluid surrounding the bubble for Re=318.63,β=22.89,γ=103, and at the time t=293.

**Figure 13 polymers-14-04259-f013:**
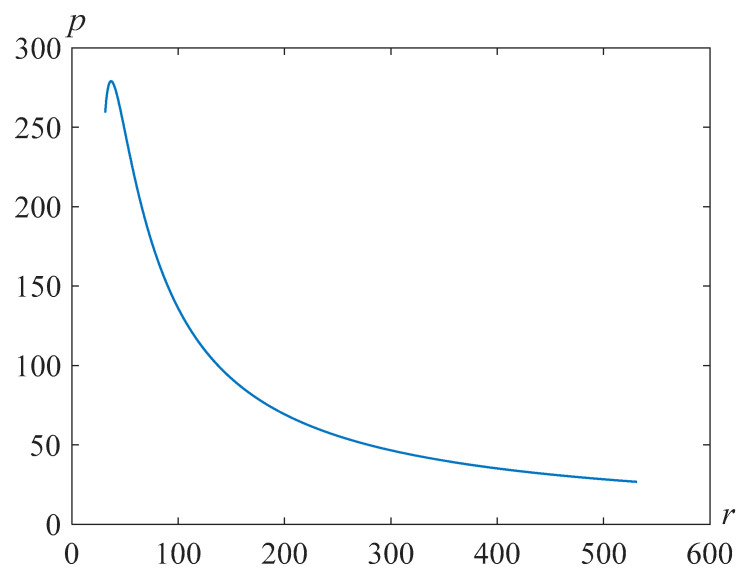
Pressure distribution in the fluid surrounding the bubble for Re=318.63,β=22.89,γ=100, and at the time t=273.

**Figure 14 polymers-14-04259-f014:**
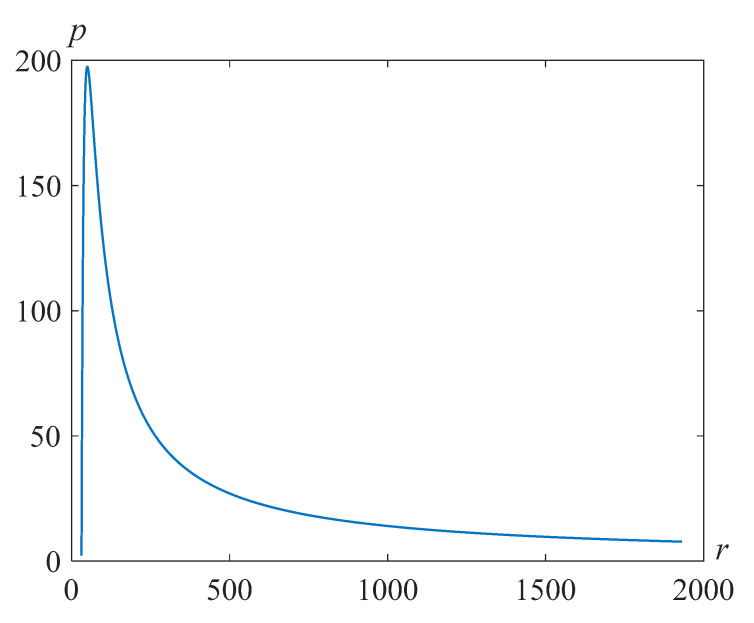
Pressure distribution in the fluid surrounding the bubble for Re=318.63,β=22.89,γ=0, and at the time t=271.

**Figure 15 polymers-14-04259-f015:**
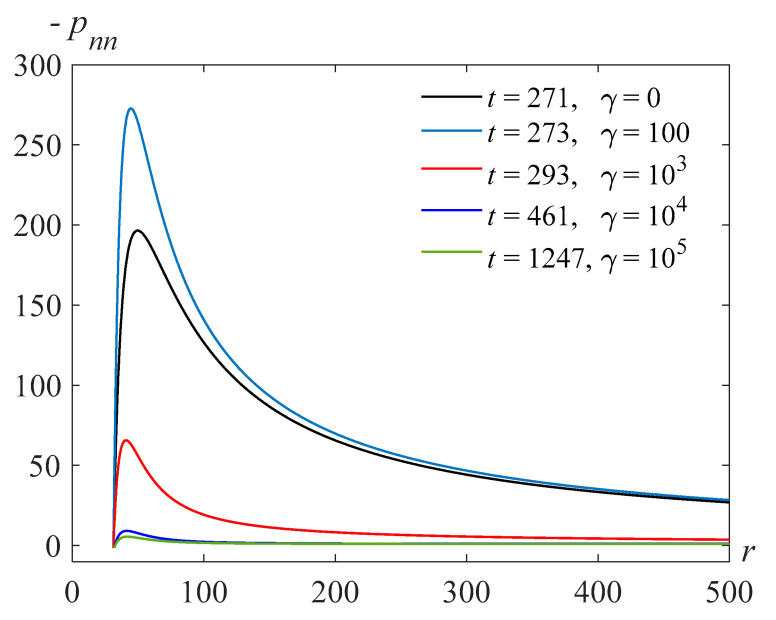
Normal stress in the fluid surrounding the bubble for Re=318.63,β=22.89, and various t and γ.

## Data Availability

Not applicable.

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
