# Peer review of "The Problem of Filling a Spherical Cavity in an Aqueous Solution of Polymers"

_polymers, 2022, doi:10.3390/polym14204259_

Round 1
Reviewer 1 Report
This article is recommended for publication after Major revision.
Manuscript ID: polymers-1918516
Type of manuscript: Article
Title: Problem of filling of a spherical cavity in an aqueous solutions of
polymers
Authors: Oxana A. Frolovskaya, Vladislav V. Pukhnachev *
Submitted to section: Polymer Physics and Theory,
I have checked the manuscript in detail. In this interesting manuscript, the analysis is all right and the
results seem correct. The manuscript is well-written and well-structured. It can be accepted for
publication but some questions need to be addressed first.
- Motivation of the study must be included in abstract.
- Updated literature regrading the present study should be addressed
- Justify how your results are accurate.
- All equations are not numbered in the draft.
- Units with all fluid properties are not properly mentioned.
- How Re is defined on page 4.
- Compare the results of different models that you have used in your study
- Remove the grammatical errors like, The law of behavior of the medium given…, second grade fluid equations allow an a priori estimate….., . It is this situation that is considered …
Recommendation: I would recommend that authors do a thorough revision considering comments prior to a re-submission before this paper can be considered for publication in Polymers
Author Response
Response to Reviewer 1 Comments
Point 1: Motivation of the study must be included in abstract.
Response 1: Motivation of the study is included in abstract.
Point 2: Updated literature regarding the present study should be addressed
Response 2: The reference to five papers and one monograph has been added. In the Introduction, we have extended the description of previous works on the subject of the paper.
Point 3: Justify how your results are accurate.
Response 3: Accuracy of results is justified by using reliable numerical methods and comparison of numerical and analytical results in limit cases.
Point 4: All equations are not numbered in the draft.
Response 4: The equations that are referenced in the text are numbered.
Point 5: Units with all fluid properties are not properly mentioned.
Response 5: The corrections are introduced.
Point 6: How Re is defined on page 4.
Response 6: The Reynolds number Re is defined as r0(p0/ρ)1/2ν-1.
Point 7: Compare the results of different models that you have used in your study.
Response 7: Comparison of the results is given in the Sections 5 (page 12) and 7 (page 16).
Point 8: Remove the grammatical errors like, The law of behavior of the medium given…, second grade fluid equations allow an a priori estimate…. . It is this situation that is considered …
Response 8: We have adopted your suggestions concerning the clarity of presentation and eliminated the grammatical errors.
In accordance with your suggestions we have substantially revised the text of the manuscript, clarified the derivation of the equations, and obtained new results.

Reviewer 2 Report
Manuscript ID: Polymers-1918516
Author studied “Problem of filling of a spherical cavity in an aqueous solutions of polymers”. After reading the manuscript, I suggest a minor revision with the below comments and suggestions.
Line 2: Title….an aqueous solutions of polymers => an aqueous solution of polymers (Check English)
Line 13: the cases of an inviscid and viscous fluid. => (Check English)
Line 19, 20: “The addition of soluble polymers to water practically does not change the viscosity, density, thermal conductivity, and other properties of the solution ”
Comments: This is not acceptable because the viscosity is dependent on the polymer concentration. Hence, if authors want to keep this sentence, some specifications are needed, e.g., extremely diluted conditions, etc. In this case, please specify the range of concentration.
Line 56: polyacrylamide => polyacrylamide (PAM) with the formula (-CH2CHCONH2-)
Line 120: polyacrylamide => PAM
Line 120-121: concentration of about 10-4 => (dimension needed)
All Figures: x-axis and y-axis’ title: (1) Please do not bury the title inside of a graph scale, (2) If there is a dimension, please add it. For example, t => t (sec)
Line 183: Figures 2, 3 => Figures 2 and 3
Finally, Authors submitted this manuscript to “Polymers” journal. However, I could not find any polymer story. Hence, author may add information such as “the lists of water soluble polymers and their physical properties (e.g., solution viscosity at the same concentration; glass transition temperature, etc.)” using a Table. Otherwise, this article is simply dealing with a fluid dynamics, indicating it is not much interesting to the polymer society.
Author Response
Response to Reviewer 2 Comments
Point 1: Line 2: Title….an aqueous solutions of polymers => an aqueous solution of polymers (Check English)
Response 1: The correction is introduced.
Point 2: Line 13: the cases of an inviscid and viscous fluid. => (Check English)
Response 2: We have eliminated the grammatical errors.
Point 3: Line 19, 20: “The addition of soluble polymers to water practically does not change the viscosity, density, thermal conductivity, and other properties of the solution ”
Comments: This is not acceptable because the viscosity is dependent on the polymer concentration. Hence, if authors want to keep this sentence, some specifications are needed, e.g., extremely diluted conditions, etc. In this case, please specify the range of concentration.
Response 3: We have emphasized that the present paper is devoted solely to the investigation of the problem under consideration in weakly concentrated polymer solution. We consider models describing the motion of a fluid with small polymer additives (with a concentration of 10-2 percent). In these models, the shear viscosity is assumed to be constant [23, 28].
Point 4: Line 56: polyacrylamide => polyacrylamide (PAM) with the formula (-CH2CHCONH2-)
Response 4: The correction is introduced.
Point 5: Line 120: polyacrylamide => PAM
Response 5: The correction is introduced.
Point 6: Line 120-121: concentration of about 10-4 => (dimension needed)
Response 6: The dimension is added.
Point 7: All Figures: x-axis and y-axis’ title: (1) Please do not bury the title inside of a graph scale, (2) If there is a dimension, please add it. For example, t => t (sec)
Response 7: The correction is introduced. In all figures, plots are presented in dimensionless variables.
Point 8: Line 183: Figures 2, 3 => Figures 2 and 3
Response 8: The correction is introduced.
Point 9: Finally, Authors submitted this manuscript to “Polymers” journal. However, I could not find any polymer story. Hence, author may add information such as “the lists of water soluble polymers and their physical properties (e.g., solution viscosity at the same concentration; glass transition temperature, etc.)” using a Table. Otherwise, this article is simply dealing with a fluid dynamics, indicating it is not much interesting to the polymer society.
Response 9: Some properties are measurable, for example the fluid relaxation time. Relaxation viscosity coefficient κ characterizes the properties of a liquid. Its value depends on the type of polymer and its concentration. The authors of the model [23] did not provide its characteristic values, though one can assume that they are sufficiently small. In our previous work [26] we discussed the possibility of experimental determination of this parameter.
In accordance with your suggestions we have substantially revised the text of the manuscript, clarified the derivation of the equations, and obtained new results.

Round 2
Reviewer 1 Report
The present form of the article is recommended for the publication